# The Multifarious Applications of Copper Nanoclusters in Biosensing and Bioimaging and Their Translational Role in Early Disease Detection

**DOI:** 10.3390/nano12030301

**Published:** 2022-01-18

**Authors:** Kumar Babu Busi, Mathangi Palanivel, Krishna Kanta Ghosh, Writoban Basu Ball, Balázs Gulyás, Parasuraman Padmanabhan, Sabyasachi Chakrabortty

**Affiliations:** 1Department of Chemistry, School of Engineering and Sciences, SRM University AP Andhra Pradesh, Gunntur, Andhra Pradesh 522502, India; busi_kumar@srmap.edu.in; 2Lee Kong Chian School of Medicine, Nanyang Technological University Singapore, 59 Nanyang Drive, Singapore 636921, Singapore; PMAT0001@e.ntu.edu.sg (M.P.); gkkanta@ntu.edu.sg (K.K.G.); balazs.gulyas@ntu.edu.sg (B.G.); 3Department of Biological Sciences, School of Engineering and Sciences, SRM University AP Andhra Pradesh, Guntur, Andhra Pradesh 522502, India; writoban.b@srmap.edu.in

**Keywords:** noble metals, copper nanoclusters, fluorescence, diagnostics, theranostics, bioimaging, biosensing

## Abstract

Nanoclusters possess an ultrasmall size, amongst other favorable attributes, such as a high fluorescence and long-term colloidal stability, and consequently, they carry several advantages when applied in biological systems for use in diagnosis and therapy. Particularly, the early diagnosis of diseases may be facilitated by the right combination of bioimaging modalities and suitable probes. Amongst several metallic nanoclusters, copper nanoclusters (Cu NCs) present advantages over gold or silver NCs, owing to their several advantages, such as high yield, raw abundance, low cost, and presence as an important trace element in biological systems. Additionally, their usage in diagnostics and therapeutic modalities is emerging. As a result, the fluorescent properties of Cu NCs are exploited for use in optical imaging technology, which is the most commonly used research tool in the field of biomedicine. Optical imaging technology presents a myriad of advantages over other bioimaging technologies, which are discussed in this review, and has a promising future, particularly in early cancer diagnosis and imaging-guided treatment. Furthermore, we have consolidated, to the best of our knowledge, the recent trends and applications of copper nanoclusters (Cu NCs), a class of metal nanoclusters that have been gaining much traction as ideal bioimaging probes, in this review. The potential modes in which the Cu NCs are used for bioimaging purposes (e.g., as a fluorescence, magnetic resonance imaging (MRI), two-photon imaging probe) are firstly delineated, followed by their applications as biosensors and bioimaging probes, with a focus on disease detection.

## 1. Introduction

While the early diagnosis of diseases is a critical task for all humans because it allows diseases to be treated while they are still in their early stages, it nevertheless is a challenging one. In particular, some diseases such as cancer, Alzheimer’s and Parkinson’s diseases do not manifest telltale signs and symptoms until they have progressed to a more advanced stage [1]. There is thus a pressing need to investigate biomarkers, imaging modalities and probes that can operate synergistically in detecting disease during its nascent stages. Such early diagnosis of disease not only prepares patients adequately for the prognosis of disease, but may even be able to circumvent the progression of disease to its advanced stages by virtue of protective treatments. Bioimaging plays an instrumental role in disease detection, during both its early and later stages. Bioimaging refers to non-invasive technologies for the visualization of biological processes in real time, which includes the study of subcellular structures, whole cells, tissues, and multicellular organisms [2,3,4,5]. It uses a variety of imaging sources, such as light, fluorescence, electrons, MRI, X-ray, ultrasound and positrons [6]. However, bioimaging modalities would not be able to provide useful information about a patient’s disease status quo without the support of probes.

Nanoparticles have played an instrumental role in complementing the usage of biomedical imaging techniques [7]. Biomedical imaging techniques such as MRI, computed tomography, and ultrasound have advanced significantly in recent years, and played critical roles in clinical cancer management. Molecular imaging (MI) is a non-invasive imaging technique used to monitor biological processes at the cellular and subcellular levels. MI achieves these goals by utilizing targeted imaging agents that can bind targets of interest with high specificity and report on associated abnormalities, a task that conventional imaging techniques cannot perform. In this regard, MI holds great promise as a potential therapeutic tool for cancer early detection [8]. Despite this, the clinical applications of targeted imaging agents are limited due to their inability to cross biological barriers within the body. The use of nanoparticles has enabled us to overcome these constraints.

In recent studies, metal nanoclusters (NCs) have garnered enormous interest for use as probes owing to their fascinating characteristics, such as their ultrasmall size, strong fluorescence, lasting colloidal stability, low toxicity, biocompatibility, and strong photostability [9]. Amongst all the metal NCs, copper (Cu) possesses favorable properties, such as excellent conductivity, abundance, and a relatively low cost compared to gold, silver and platinum (Au, Ag, Pt) [10]. Therefore, we have in this review opted to address the fluorescence characteristics of Cu NCs employed in bioimaging applications with numerous imaging sources. The following sections detail the different modes in which Cu NCs probes may be applied in potential biosensing and bioimaging applications.

## 2. Potential Modes of Copper Nanocluster Usage in Bioimaging

### 2.1. Advantages of Copper over Other Metals

Amongst the group 11 elements, copper, silver, gold and platinum (Cu, Ag, Au, Pt) are elements that possess a higher enthalpy of sublimation and ionization energy, which means they demonstrate a noble character [11]. These noble metals are resistant to corrosion and oxidation, and are inert in the face of non-oxidizing acids, allowing researchers to work on these metals. Amongst all the metals, Au and Ag have been extensively researched in terms of their size-controlled synthesis, intermetallic reaction mechanisms, structural characterization, ligand-induced optical behavior, properties, biological applications, and catalytic performance [12]. Due to their comparatively simple chemical composition, Cu NCs are easily manipulated for optimization as compared to multicomponent systems (up-conversion nanoparticles and liposomes Cu). Cu NCs have gained much attention over Ag or Au NCs because of (i) their high yield in mild synthetic conditions, and (ii) the abundance and low cost of Cu, which offers practical possibilities for large-scale nanotechnology applications, and in day-to-day clinical settings, in providing affordable health care. Finally, in contrast to gold, silver and platinum, Cu is an essential trace element in the human body, and therefore, an excess of Cu can be effectively removed by physiological processes [13]. These give Cu NCs an additional advantage when used in diagnostics and therapy. Nevertheless, they attention being paid to the development of Cu is relatively limited owing to the difficulty of preparing small clusters and their susceptibility to oxidation.

### 2.2. As a Fluorescence (Visible and Near Infrared Region) Probe

Optical imaging technology has become a critical research tool in the field of biomedicine, thanks to its ability to visualize biomolecules, cells, tissues, and living organisms in real time and in three dimensions [12]. Compared with other technologies, optical imaging technology has a number of advantages, including the fact that it is non-invasive and safe, has excellent spatial resolution and visualization capabilities, produces a great volume of data quickly, and is a low-cost method. It has been employed in biomolecular imaging, drug distribution, metabolic tracking, disease detection, and diagnostics. Optical imaging technology has a promising future, particularly in early cancer diagnosis and imaging-guided treatment [10].

The illumination of fluorescence is the radiative emission that arises from the electronic transition between two distinct energy states within a luminescent material [14]. As the sizes of bulk metals approach the nanoscale, they exhibit molecule-like properties by nearing the fermi wavelength of an electron through a breakdown of uninterrupted energy bands into discrete energy levels [15,16]. This is responsible for unique physiochemical and molecular properties, such as strong absorption, a large Stokes shift, larger surface-to-volume ratio and bright fluorescence [17].

Over the last decade, noble metals such as Pt, Ag, Cu and Au have been widely examined due to their ultrasmall nature, with a core size of less than 2 nm, which exhibits the property of luminescence. The fluorescence of metal NCs may be due to the transition of electrons between completely filled d-bands and the states above the fermi level, or the electron shifts between the highest-occupied and lowest-unoccupied molecular orbitals (HOMO and LUMO, respectively). On rare occasions, dual emission has been observed in Cu NCs, whereby the high emission energy is assigned to the transition from excited levels in the sp- to the d-band [18]. On the other hand, the low emission energy is due to the intraband shift from the excited states within the sp-band. A combined application of interband and intraband transitions was demonstrated by Wei et al., whereby blue-emitting Cu NCs were generated through a facile one-pot method in which the solvents, hexane, toluene and chloroform were applied in a traditional wet chemical method [19]. Their results confirmed that the small-sized Cu*_n_* (*n* ≤ 8) clusters demonstrated dual emissions at 423 nm and 593 nm, akin to those in Au NC complexes, whereby the 423 nm emission peak was due to an interband shift from excited states in the sp-band to the d-band, and the 593 nm emission peak was due to intraband transition within the sp-band. Following this, red-emitting Cu NCs were prepared by the same group to assess their influencing properties at the nanoscale level in biological systems.

In biological organisms, red or infrared (IR) fluorescence may be utilized to avoid interference with the innate fluorescence of living species, as many luminous compounds deliver other visible colors, such as blue or green [20]. Furthermore, red or IR-emitting fluorophores can easily and more effectively penetrate living organisms than blue and green fluorophores, which could be advantageous for deep tissue imaging applications. Wang et al. reported a facile production of stable and fluorescent Cu NCs, which was protein-directed, using bovine serum albumin (BSA) as a stabilizing agent [21] or template. Hydrazine hydrate (N_2_H_4_•H_2_O) was utilized as a mild reducing agent to obtain red fluorescence peaking at 620 nm with a 4.1% quantum yield (QY), as shown in Figure 1a. In addition, these as-synthesized Cu NCs were employed as probes for cellular imaging (CAL-27) due to their distinct features, such as better water solubility, red fluorescence, surface activity, biocompatibility, etc. Additionally, there are multiple factors influencing the fluorescent properties of Cu NCs with respect to enhancement or quenching, which will be discussed in the following section.

#### 2.2.1. Optical Windows in Fluorescence Imaging

Fluorescence is the emission of light by a substance following the absorption of photons or other electromagnetic radiation of a different wavelength. In most cases, the emitted light has a longer wavelength, and therefore lower energy, than the absorbed radiation. However, when the absorbed electromagnetic radiation is intense, it is possible for one molecule to absorb two photons; this two-photon absorption may lead to the emission of radiation with a shorter wavelength than the absorbed radiation [22]. Fluorescence imaging has evolved into a novel and promising tool for in vivo imaging, as it can achieve real-time sub-cellular resolution imaging, and hence be widely used in the field of biological and medical detection and treatment. However, it is unable to highlight biological complexities in deep tissues due to the limited imaging depth (1–2 mm) and self-fluorescence background of tissue released in the visible range (400–700 nm). Because it minimizes NIR absorption and scattering from blood and water in organisms, the conventional near-infrared wavelength (NIR-I, 650–950 nm) is considered the first biological window [23]. The penetration of NIR fluorescence in bioimaging is greater than that of visible light. In fact, tissue autofluorescence (background noise) and photon scattering, which restrict tissue penetration depth, continue to disrupt NIR-I fluorescence bioimaging. The signal-to-noise ratio (SNR) of bioimaging can be greatly increased in the second near-infrared range (NIR-II, 1000–1700 nm), also known as the second biological window, according to recent experimental and computational studies. NIR-II bioimaging can investigate deep-tissue information at centimeter depths, and acquire micron-level resolution at millimeter depths, exceeding the capabilities of NIR-I fluorescence imaging. The key to fluorescence bioimaging is to use a functional or targeting contrast agent to accomplish extremely selective imaging (probe). NIR-II probes, on the other hand, have made relatively little progress. Only a few papers on NIR-II fluorescence probes, such as carbon nanotubes, Ag2S quantum dots, and organic small molecule dyes, have been published to date [24]. Among these fluorescent probes, metal NCs have attracted huge interest because of their small size (<2 nm), low toxicity, good biocompatibility and strong fluorescence. Particularly in early cancer diagnosis and imaging-guided treatments, fluorescence imaging technology has a promising future when using these probes.

#### 2.2.2. The Impact of Different Factors on Fluorescence Enhancement of Cu NCs

The ultrasmall size of Cu NCs ushers in a new era of quantum confinement, making them far more fascinating due to their fluorescence properties [25]. While the exact fluorescence mechanism of Cu NCs is not yet completely understood because of their sensitive nature towards their chemical environment, it is apparent that, in order to achieve highly stable and strongly fluorescent Cu NCs, they should be safeguarded from aggregation and oxidation. Hence, the fine tuning of the following factors can considerably improve the Cu NCs’ stability and fluorescence: (a) the cluster size of the metal core, (b) the presence of surface engineering with suitable ligands to protect the Cu NCs from oxidation, (c) other synthetic parameters such as pH, the composition of chemicals, reaction temperature, reaction time and concentration of reducing agents, etc., which can greatly affect the fluorescence properties. These will be discussed in detail in the next sections.

**Figure 1 nanomaterials-12-00301-f001:**
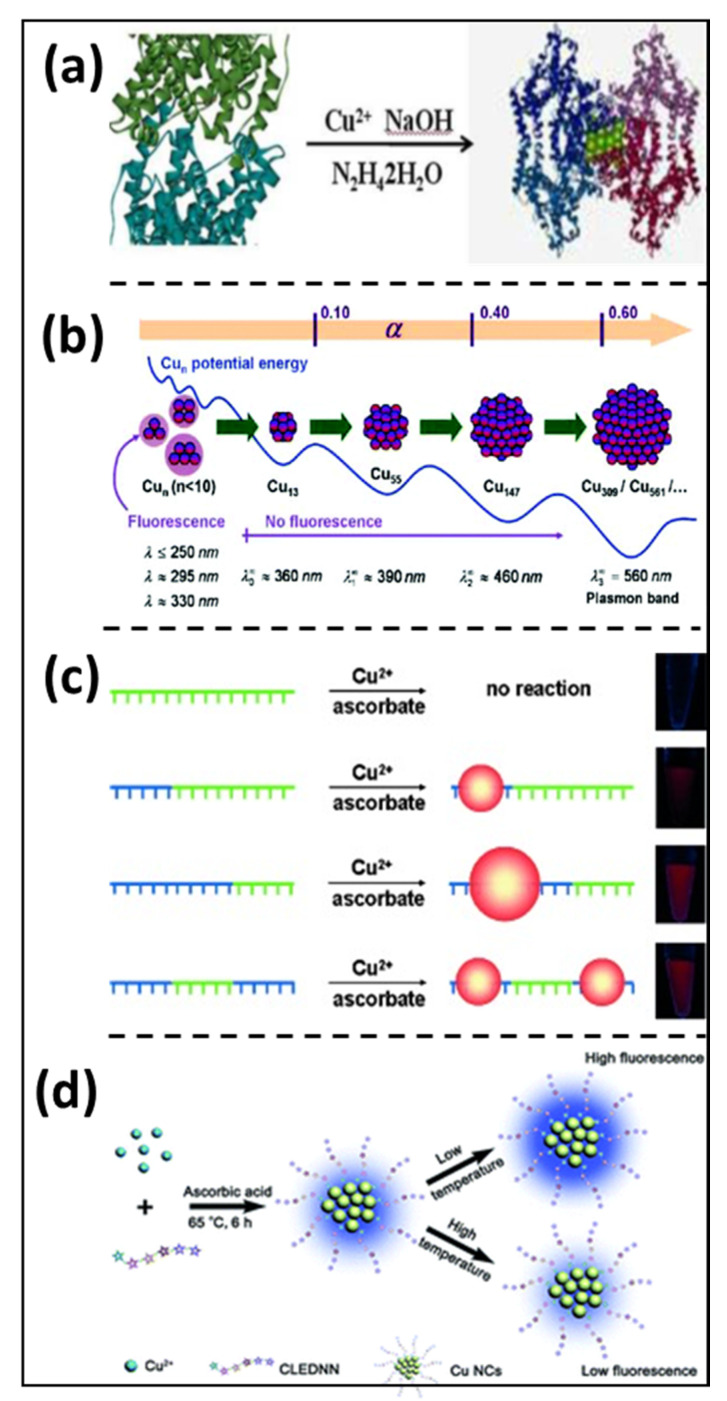
Cu NCs’ photoluminescence properties can be drastically varied with (**a**) a facile route to prepare aqueous phase protein-derived red fluorescence-emitting copper nanoclusters for cell imaging. (Reprinted with permission from ref. [21]. Copyright 2014 Royal Society of Chemistry.) (**b**) By increasing the molar percentage of NaBH_4_, the size of copper nanoclusters evolves schematically. Big particles with n 309 show a characteristic plasmon resonance band, whereas clusters with n 13 appear fluorescent. (Reprinted with permission from ref. [26]. Copyright 2009 American Chemical Society.) (**c**) The controlled reduction in DNA-complexed Cu^2+^ ions to produce nanoparticles on preselected ssDNA regions. Cu-ssDNA (blue) is a type of ssDNA that can be employed as a template to make CuNPs. The length of the CussDNA determines the size of the CuNPs produced. Cu-ssDNA (green) is ssDNA that cannot make CuNPs, which are employed to connect nanoparticles in complicated nanostructures. (Reprinted with permission from ref. [27]. Copyright 2013 Angewandte Chemie.) (**d**) Schematic representation of the creation procedure of Cu NCs with temperature-dependent fluorescence. (Reprinted with permission from ref. [28]. Copyright 2014 Royal Society of Chemistry).

##### Effect of Cluster Size

Earlier studies have already affirmed that the photoluminescence property mostly depends on the cluster core size [29]. At the atomic level, the choice of precursors and reaction parameters is crucial in altering the size of Cu NCs, which have a quantum size effect. Due to this, the continuous conduction bands are broken down into discrete electronic states in Cu NCs (with sizes less than 2 nm). In 2009, Vazquez-Vazquez’s group explained the fluorescence property of Cu NCs by controlling the concentration of the reducing agent used [26]. Precisely, when less than 10 atoms were used, Cu NCs exhibited high fluorescence intensity, as shown in Figure 1b. In contrast, the emission intensity of Cu NCs diminished when the reducing agent was increased in concentration, resulting in the generation of bigger-sized particles with a red shift of the UV-visible absorption band. Recently, Wang et al. also showed that the fluorescence of Cu NCs is highly proportionate to their size, whereby an increase in the average size of NCs from 1.8 nm to 3.5 nm caused the Cu NCs’ PL intensity to display a prominent red shift (~130 nm) from blue to orange emissions [30].

##### Effect of Ligand

In addition to the size of the metal core, ligands also have crucial importance in tuning the optical and chemical properties of Cu NCs. Surface protective ligands play a pivotal function in controlling the optical characteristics, and predominantly in fluorescence enhancement. The effect of templates/polymers with a predetermined structure is to bind to Cu^2+^ ions first, which then become reduced into Cu atoms and clusters on the template, forming Cu NCs. Steric protection by the template avoids the aggregation of the NCs, or their decomposition in solution [15]. Additionally, the ligands have a substantial impact on the stability of the NCs by controlling metal core size and offering protection from aggregation. Co-ordination between the metal core and ligand typically results in a ligand-to-metal charge transfer (LMCT) or ligand-to-metal–metal charge transfer (LMMCT), and consequently radiative relaxation could occur resulting from the triplet state of the metal center. To improve the fluorescence emission of Cu NCs, both the capability of the ligand to give away electrons and the electro-positivity of the metal core could be considered.

Wang et al. reported the generation of Cu NCs with different single-stranded DNA (ssDNA) scaffolds, including poly(adenine) (poly A), poly(thymine) (poly T), poly(cytosine) (poly C) and poly(guanine) (poly G), as shown in Figure 1c. They discovered that only the template based on poly T caused the creation of fluorescent Cu NCs, with PL emission and excitation peaks at wavelengths of 600 nm and 340 nm, respectively. Both the PL QY and the size of the Cu NCs could be altered by varying the length of the poly T sequence, whereby longer stretches of poly T trigger the development of larger clusters with a higher PL QY [27]. A polyvinylpyrrolidone (PVP) ligand can also affect the formation of Cu NCs by wrapping onto the surface of NCs. It can provide binding sites for Cu^2+^ based on the interaction between single nitrogen atoms and the empty orbitals of Cu^2+^. Wang et al. reported that the aqueous synthesis of PVP supported Cu NCs, which emanate in the blue region spectrum with a PL QY of 8%. The synthesized Cu NCs’ PL QY improved greatly to 27% after treatment with glutathione (GSH) [31]. Polyethylene glycol (PEG) is also a promising stabilizing agent for Cu NCs, as it can improve the PL stability of NCs, and the PL QY might be enhanced. Differently structured ligands can affect the formation of clusters by providing multiple functional groups, and they can function as good stabilizing and reducing agents as well [32]. These results confirm that ligands with functional groups abundant in electrons have a strong electron-donating ability, which will enormously enhance the PL QY of Cu NCs. Additionally, in terms of PL QY enhancement, the reaction temperature affects the rate of reaction and growth kinetics of particles in the reaction medium.

##### Effect of Reaction Temperature

Reaction temperature is a key consideration in the synthesis of metal NCs or any kind of nanoparticle (NP), as the growth, size and monodispersity of the NPs completely depends on the reaction temperature. For example, Xie et al. reported that Au NCs prepared by commercially available BSA protein can in themselves act as a stabilizing and reducing agents because of the multiple functional groups, such as −OH, −SH, NH_2_ and −COOH [33], present in the protein backbone. As the prepared BSA templated Au NCs produced a red emission peak centered at around 640 nm via the simple one-pot chemical reduction method, they observed reasonable reduction kinetics at a physiological temperature of 37 °C, with a high QY of ~6%, and reaction was completed in 12 h. When the reaction temperature was maintained at 100 °C, the reaction was achieved within minutes with a low QY (~0.5%). Similarly, Huang et al. reported a facile eco-approach for the fabrication of fluorescent Cu NCs using an artificial peptide with amino acid sequence CLEDNN as a template [28]. The Cu NCs possessed a high QY of around 7.3% at a 65 °C reaction temperature. Owing to the temperature-dependent fluorescence properties that ensued due to the modification, the blue-emitting Cu NCs were able to sense the intracellular temperature, as shown in Figure 1d.

#### 2.2.3. Aggregation-Induced Emission

Aggregation-induced emission (AIE) is a novel photo-physical aspect in which weakly luminescent or non-luminescent nanomaterials will become extremely luminescent after aggregation either in the colloidal state or in the solid state [34,35,36,37]. The AIE phenomenon was first discovered by Tang et al. to achieve a high PL QY in organic molecules of 1-methyl-1,2,3,4,5-pentaphenylsilole [38]. AIE’s characteristics in metal NCs consist of high PL QYs (commonly ranging from 10 to 50%), long excited state lifetimes and large Stokes shifts. Among the various ligand-stabilized metal NCs, those surface-capped with GSH, 1-dodecanethiol, cysteine and penicillamine are the most frequently described agents [39]. In particular, Cu NCs stabilized by thiol ligands were found to have amended AIE traits, resulting in the enrichment of the PL QY by more than 10 times.

Although metal NCs require further studies on their AIE properties in order to ground a complete understanding of their exact mechanism of action, this mechanism has been generally associated with the limitations of the intramolecular motion of capping ligands in the aggregation states, blocking the non-radiative pathways, and subsequently activating the radiative decay, resulting in the PL enhancement of metal NCs [40]. As in Figure 2, the emission of isolated metal NCs exhibits fluorescence (transition from S_1_ to S_0_), while aggregated NCs emit phosphorescence (transition from T_1_ to S_0_). In addition, the colors of most AIEs of aggregated NCs appear in the red and near-IR spectral regions, thus they can be useful in bioimaging applications. The AIE effect in Cu NCs can be activated by post-synthetic modifications with various parameters, such as solvents, pH, or the addition of various ions to stimulate their clustering.

##### Effect of Solvent

The AIE properties of Cu NCs can be precisely regulated or manipulated by virtue of polar solvents, such as ethanol, methanol or water, for hydrophobic Cu NCs [41]. Maity et al. described that the addition of ethanol to GSH-Cu NCs significantly improved their fluorescence emission properties [42]. Cu NCs containing negatively charged surfaces due to ligands (in amino acids, peptides, proteins, etc.) comprising various functional groups, such as COOH and OH, provide better solubility and stability due to a thick surrounding hydration layer being formed in aqueous medium. As a result of increasing the concentration of less polar solvents such as ethanol in the aqueous medium, the hydration layer around the Cu NCs shrinks, neutralizing the surface negative charges (ζ). This brings about a reduced stability and closer proximity of the NCs because of the enrichment of intra- and inter-molecular Cu(I)–Cu(I) cuprophilic interactions. Therefore, the interaction between ligand shell and metal core, together with the limitations in ligand vibrations and rotations in the clustered state, led to a drastic change in emission characteristics; the non-radiative pathways were reduced, whereas radiative ones were improved. As shown in Figure 3a, Maity et al. reported the same phenomena, whereby with the addition of 90% ethanol, the weakly orange color-emissive Cu_34–32_(SG)_16-13_ NCs with QY 0.03% were considerably increased up to 36-fold [42].

Ling et al. studied the AIE property of polyethyleneimine (PEI)-encapsulated Cu NCs dispersed in 12 disparate organic solvents [43]. They confirmed that the optical characteristics of PEI-Cu NCs in alcohol solvents (including methanol, ethanol, isopropanol, n- propanol, ethylene glycol and n- butyl alcohol) exhibited similar properties to those in water. Among them, tetrahydrofuran (THF) had a greater potential for making hydrogen linkages as compared with water. However, PEI-Cu NCs exhibited a blue shift of the emission peaks in THF and 1,4-dioxane, and their absorption spectra had been altered drastically. Furthermore, the optical alterations of PEI-Cu NCs in THF confirmed that the fluorescence intensity can be enhanced greatly in the aggregated state. Liu et al. reported GSH-derived Cu NCs with a strong red fluorescence by controlling Cu–Cu interactions and hydrogen bonding, which resulted in a PL QY of approximately 3%. Cu NCs with red fluorescence may display a strong red shift by emitting strong orange emissions in the solid form rotary evaporation and lyophilization processes, via manipulating the Cu–Cu and hydrogen-linkage interactions. Cu NCs can form a gel with bright orange fluorescence in DMF, according to the study of several polar solvent-induced aggregations [44].

##### Effect of pH

Another key aspect to consider when controlling the PL properties of AIE is the pH of the environment. The production of hydrogen bonds or protonation/deprotonation activities, which cause major changes in the PL characteristics of NCs, are thought to be responsible for the AIE amplification. Su et al. reported a simple pH-guided strategy to assemble proteins and Cu NCs into hybrid nanostructures soluble in water, along with good stability and bright emission from AIE [45]. The protein-conjugated Cu NCs are known to be minutely responsive to pH changes due to alterations in the structure of the protein backbone, as shown in Figure 3b. Using L-cysteine as both a capping and reducing agent, Cu NCs generated insoluble macroscopic aggregates with red color emissions (620 nm, QY 5.4%) at pH 3.0. These aggregates were soluble at pH < 1.5 or pH > 4.0 with a weak emission. The addition of the graphene oxide (GOx) enzyme or BSA protein induced an enhancement in PL QY to 6.3% due to the formation of larger aggregates. The AIE improvement in cysteine-stabilized Cu NCs was due to the production of hydrogen bonds between the carboxyl and amine terminals under low pH conditions [46]. The emission efficiency improved vastly in aggregated Cu NCs for two crucial reasons: (1) the increased cuprophilic interactions (Cu(I)–Cu(I)) contributing to more radiative relaxation of the excited state; (2) the restriction of intermolecular and intramolecular vibrations and rotation, moving the nonradiative pathway away from the excited state’s relaxation dynamics.

**Figure 3 nanomaterials-12-00301-f003:**
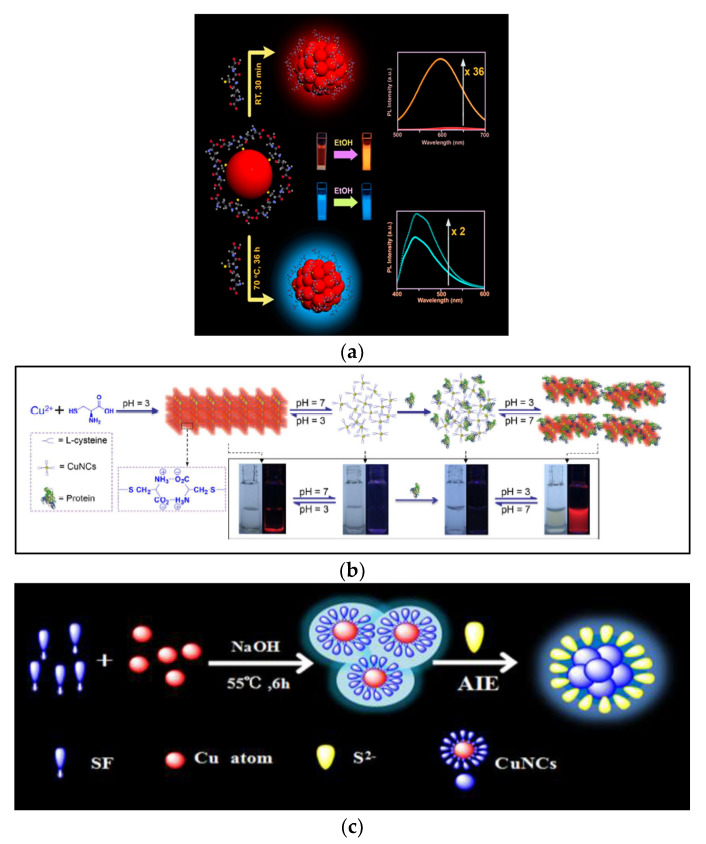
(**a**) Using a top-down approach, a schematic diagram of the GSH-Cu NCs is shown. When the Cu_25_(SG)_20_ NCs are exposed to ethanol for 36 h at 70 °C, they emit a strong blue fluorescence. The enhancement of weak orange emissive Cu_34–32_(SG)_16-13_ NCs with QY rose 36-fold when 90% ethanol was added. (Reprinted with permission from ref. [42]. Copyright 2019 American Chemical Society.) (**b**) Water-soluble protein/Cu NCs hybrid nanostructures can be made using a pH-guided approach. (Reprinted with permission from ref. [45]. Copyright 2017 American Chemical Society.) (**c**) With the great assistance of S^2−^ in aqueous solution, the silk fibroin-coated copper nanoclusters were successfully proved to have unique aggregation/assembly-driven emission enhancement features. (Reprinted with permission from ref. [47]. Copyright 2019 Sensors and Actuators B: Chemical.)

##### Effect of Ion

Metal cations have a synergistic effect on the periphery of Cu NCs, which can enhance the emission efficiency by up to several orders of magnitude in the aggregated state. In general, clustering triggered by metal cations may be ascribed to the neutralization of charges on the surface of Cu NCs or the development of new linkages due to enhanced electron affinity. Chen et al. explored the emission behavior of water-soluble (±)-α-Lipoic acid (DHLA)-protected Cu NCs (with QY 2.8%) in the presence of various metal cations (inclusive of Co^2+^, Mg^2+,^ NH^4+^, Mn^2+^, Pb^2+^, Cu^2+^, Fe^3+^, Ca^2+^, Hg^2+^, Al^3+^, Ag^+^ and Zn^2+^) to investigate the emission enhancement. Among these, the amphoteric metal cations, such as Zn^2+^ and Al^3+^, exhibited doubling in the fluorescence due to their combination with OH groups on Cu NC surfaces to promote the formation of larger aggregates [48]. Zhang et al. demonstrated that silk fibroin (SF)-derived Cu NCs produce unique AIE enhancement properties with the aid of S2- in aqueous solution, as depicted in Figure 3c. The interaction between S^2−^ and the core SF-Cu NCs can induce ultrasmall Cu NCs of 2.8 nm to precisely assemble into large aggregates of uniform rod-shaped fluorescent nanoparticles, with the dimensions of (47 ± 2) × (15 ± 2) nm. The absolute PL QY also improved from 1.6% to 4.9%. This might be expanded to the identification of S^2−^, with a detection limit of 0.286 μM, in a linear concentration range of 5–100 μM [47].

#### 2.2.4. Different Factors Impact Cu NCs Fluorescence Quenching

Fluorescence quenching is a phenomenon that decreases the intensity of emitted light from fluorophores [49,50]. When light is absorbed by a molecule, the electrons in its respective atoms are excited and elevated to a higher energy level. These excited electrons are restored to the lowest energy state by dissipating energy in the form of heat or radiation, because they are unstable at higher energy levels. Fluorescence quenching arises in two ways, static quenching and dynamic quenching. The development of a complex between fluorophore and the quenching molecule causes static quenching, and the fluorophore become nonfluorescent as a result. In contrast, dynamic quenching happens via the interaction between two light-sensitive molecules; a donor and an acceptor. The donor fluorophore can transfer energy to the acceptor molecule, which can either emit light or absorb it completely [51,52].

In the last few years, Cu NCs have gained considerable attraction as a fluorescence diagnostic nanoprobe due to their low cost and abundance. Recently, Shao et al. synthesized red-emitting Cu NCs at 627 nm using dithiothreitol (DTT) as a surface capping and reducing agent at room temperature, as the DTT- Cu NCs could be utilized as a unique fluorescent nanoprobe due to their remarkable sensing of cobalt (II) ions. The detection limit can reach a low concentration of 25 nM with cobalt (II) via the fluorescence quenching mechanism [53]. Feng et al. reported trypsin-mediated Cu NCs with blue0 and yellow-emitting NCs under acidic and basic conditions. Their experimental results confirm the fluorescence-quenching mechanism for the detection of mercury (Hg^2+^) ions, with a detection limit of 30 nM [54]. Taken together, fluorescent quenching may be used as a novel diagnostic method with the help of Cu NCs.

### 2.3. Ratiometric Fluorescence Probe

The ratiometric fluorescence method detects changes in the local environment by comparing the intensity of two emission wavelengths [55]. Typically, the probe is more sensitive to environmental factors such as pH, ion concentration, viscosity and solvent polarity. Nonetheless, fluorescent sensors dependent on the single-color emission signals are subjected to interference with the analyte. Ratiometric fluorescent probes also possess robust anti-interference abilities since they offer innate auto-adjustment for more reliable quantification by computing the intensity ratio of two fluorescent signals [56,57]. To date, various fluorescent nanomaterial probes have been utilized to construct efficient ratiometric sensing platforms, including semiconductor QDs [58,59,60], carbon dots [17,61], organic polymers, metal nanoclusters and small molecules. Amongst them, Cu NCs have emerged as a novel class of fluorescent nanomaterials designed for sensing and bioimaging because that are easy to make, water-soluble, nontoxic, possess a large Stokes shift, are biocompatible and are environmentally friendly [62].

Liu et al. reported that a Cu NC-based fluorescent probe was appropriate for the real-time biosensing and bioimaging of cytoplasmic Ca^2+^ in neuronal cells [63]. The Cu NC probe displayed two independent emission peaks with good water solubility at 590 nm and 690 nm. It was observed that the green emission intensity was significantly enhanced as the Ca^2+^ concentrations increased, while red-emitting NCs remained constant. The fluorescent signal ratio (F_green_/F_red_) was used to measure the amount of Ca^2+^ in neuronal cells, as shown in Figure 4a–c. Furthermore, in the following two-minute incubation period, the addition of histamine boosted the green fluorescence and improved Ca^2+^ localization in certain parts of the neurons. In particular, it was discovered that O_2_^•−^-induced neuronal apoptosis might result from the Ca^2+^ overload in neuronal cells, offering a novel perspective to comprehend the mechanisms pertaining to oxidative stress. Recently, Hu et al. assessed polyvinylpyrrolidone (PVP)-altered Cu NCs that were formulated in water for monitoring ethanol. PVP-Cu NCs exhibited a ratiometric dual emission, including a strong green emission and weak blue emission, as illustrated in Figure 4d. The strong green emission of PVP-Cu NCs displayed a monodispersed state attributed to the generation of hydration encapsulation around Cu NCs. In the presence of ethanol, where the hydration shell is destroyed, Cu NCs tend to cluster, and result in strong blue emission. This transition in emission is due to the enrichment of Cu-Cu metallophilic interactions with the clustering of Cu NCs. Thus, PVP-Cu NCs can have a quick response (<1 min) and can detect ethanol ranging from 0 to 100%. Furthermore, the ratiometric fluorescent probe has been effectively adopted to determine the ethanol content in alcoholic beverages, demonstrating its great applicability in real-time analytical applications [64].

### 2.4. Magnetic Resonance Imaging Probe

MRI is a radiological diagnostic technique used to offer greater contrasted images of the spinal cord, vascular anatomy, soft tissues (such as the brain or abdomen) and the physiological processes in the body. Nanomaterials with both fluorescence and magnetic characteristics have received considerable recognition due to their utilizations in MRI, as well as fluorescence imaging (FI). MRI/FI could also serve as an enhanced imaging method for cancer detection and monitoring, because of the high spatial resolution of MRI and heightened sensitivity of FI. Several paramagnetic metal species, such as gadolinium, iron, and others, have been employed to produce contrast enhancement in MRI for T1 and T2 relaxation time-shortening. As illustrated in Figure 5a, Wang et al. reported on the generation of water-soluble Cu NCs by using GSH as a stabilizing and reducing agent via the sonochemical route [65]. The resultant GSH-Cu NCs were small in diameter (~1.4 ± 0.2 nm), able to display red luminescence (QY up to 5.6%), had paramagnetic features at normal temperatures (298 K), and showed good water dispersion properties because of the existence of carboxyl and amino groups in GSH. Due to the GSH-conferred protective layer, Cu NCs were able to exhibit excellent biocompatibility, negligible toxicity, and multifunctional peripheral chemistry. Furthermore, due to the covalent bonding between folic acid (FA) and GSH-Cu NCs, the nanoprobes could be made to be extremely specific in targeting gastric cancer HeLa cells with laser source confocal microscopy via FI. Additionally, due to their paramagnetic properties, FA-conjugated GSH-Cu NCs could operate as T1 contrast agents during MRI. Finally, these MRI/FI studies on different immortalized cell lines revealed that FA-conjugated GSH-Cu NCs can function as effective nanoprobes in cancer cell detection, as shown in Figure 5b,c [66].

### 2.5. Positron Emission Tomography Probe

Positron emission tomography (PET) is a nuclear medicine imaging technique that quantitates the metabolic activity of the body’s tissue cells. PET is most commonly used in patients with brain or heart disorders, as well as cancer, to monitor biochemical alterations in the body, like the heart muscle’s metabolism (the mechanism by which cells convert food into energy upon digestion and abosorption into the circulation). PET imaging has grown tremendously as a diagnostic imaging tool due to its acute precision of detection (to the picomolar level), which is roughly 10^6^ times greater than MRI. PET also has the merits of high temporal precision, limitless tissue penetration, and the capacity to perform a quantitative investigation of the entire body.

Gao et al. developed BSA as a scaffold to create ultrasmall chelator-free radioactive [^64^Cu] Cu NCs for PET imaging in an orthotopic lung cancer model. In this study, a BSA scaffold was preconjugated with luteinizing hormone-releasing hormone (LHRH) to construct [^64^Cu] CuNC@BSA-LHRH. Orthotopic lung cancer models were used to replicate localized primary cancer development and its immediate environment in order to test the features of [^64^Cu]CuNCs for the sensitive and precise imaging of lung cancer. The [^64^Cu]CuNC@BSA conjugated with receptor LHRH resulted in ultrasmall NCs of hydrodynamic size (3.8 ± 0.5 nm), which were shown to demonstrate a high radiolabeling stability and high uptake in the left lung inoculated orthotopic A549 tumors and kidney. In addition, these [^64^Cu]CuNCs demonstrated significant features, such as high radiolabeling stability, accelerated deposition, as well as speedy renal removal and tumor targeting characteristics. In comparison to near-infrared fluorescence (NIRF) imaging, PET imaging with [^64^Cu]Cu nanoclusters as radiotracers revealed more sensitive, precise, and deep penetration imaging of orthotopic lung cancer in vivo [67]. Heo et al. developed direct CXCR4-targeted ^64^Cu-CuNCs-FC131 with specific binding that enabled a high specific activity in triple-negative breast cancer (TNBC) imaging patient-derived xenograft mouse models and human TNBC tissues [68]. These Cu NCs can provide novel biological tools for PET molecular imaging and future clinical translation.

### 2.6. Fluorescent Lifetime Imaging Probe

Optical imaging with fluorescence microscopy is a crucial biological imaging technique for capturing high-resolution images of molecular contrast in living organisms [69]. Fluorescence lifetime imaging microscopy (FLIM), which describes the lifetime property of fluorescence, is an additional microscopic technique that has attained popularity because of its high sensitivity to the molecular environment and changes in molecular confirmation [70]. FLIM has been widely used to examine cellular metabolism through auto-fluorescent molecular imaging. The FLIM of auto-fluorescent molecules provides deeper insights into cellular health in a nondestructive manner, and is frequently used to conduct in vivo investigations. FLIM-based sensors utilize fluorescence molecules that are known to be sensitive when monitoring a variety of chemical and physical parameters, such as pH, temperature, ion concentration, viscosity, polarity, binding to macromolecules, etc. Therefore, FLIM can monitor a variety of events in cells and tissues, including disease development and treatment efficacy, using both endogenous and exogenous fluorophores. Finally, fluorescence imaging of the sub-cellular region is indeed crucial for clinical practice. In addition, in analytical research, the detection of physiologically important metal ions in the nanomolar concentration range is critical [70]. Low-toxic, biocompatible and fluorescent NCs present a significant advantage in resolving all these challenges.

In 2015, Das et al. reported blue-emitting GSH-Cu NCs, as shown in Figure 6, which exhibit high biocompatibility, high QY (~6%), and an excellent photostability that enhanced their compatibility when used in the imaging of cellular environments. For the cell survival and uptake assay, three carcinogenic cell lines were chosen: HeLa (malignant immortalized cell line originating from cervical cancer), MDAMB231 (human breast adenocarcinoma), and A549 (human prostate cancer) (human lung carcinoma). The Cu NCs concentrations ranging from 0 to 30 μM did not affect the cell growth morphology, and a further intracellular localization was observed through confocal microscopy, as depicted in Figure 7. In living organisms, iron (Fe3^+^) plays an instrumental metabolic role, but may be toxic when present in excess. Upon the successful addition Fe3^+^ ions to GSH-Cu NCs, the luminescence was quenched by a “turn off” behavior at higher concentrations [71]. GSH-Cu NCs would be useful for Fe3^+^ ion sensing with a smaller limit of detection (~25 nM). With the aid of fluorescence lifetime imaging microscopy (FLIM), the luminescent lifetime of the probes was found to be approximately 2.72 ns. In recent times, Shao et al. have prepared methionine-stabilized Cu NCs with fluorescence QY 4.7% and longer lifetime (8.3 μs). This nanoprobe has high sensitivity and selectivity towards norfloxacin, and results in fluorescence quenching in a broad linear spectrum from 0.05 to 250 μM, with a detection limit of 17 nM [72].

### 2.7. Two-Photon Imaging Probe

Two-photon imaging microscopy (TPIM) is a state-of-the-art version of confocal microscopy that uses two-NIR photons (700–1100 nm) as the excitation source, resulting in more efficient characterizations than a single UV-Vis photon [73]. This “localization of excitation” enables excitation to a very small focal volume (~1 femtoliter) to provide 3D images with a high spatial resolution, which is another distinct feature compared to one-photon excitation microscopy [74]. TPIM can penetrate deep into live tissues, which can extend up to 70 μm from the periphery of tissues with marginal impedance from auto-fluorescence because of the absorbing NIR photons (so-called favorable biological window). When compared to the uninterrupted wave laser used in confocal microscopy, the energy taken in by the tissue from the femtosecond pulse laser is substantially lower. As a result, the photobleaching of the probes and photo-induced deterioration of the biological samples are minimized. Therefore, a wide range of TP probes is required to make TPIM a more adaptable technique in biology, medicine, and other specific applications [75,76].

Noble metal nanoparticles have been used for TPIM, but their larger size may prevent them from being used for subcellular targeting [77]. Targeted metal cluster probes were thought to be a preferable alternative for the TPIM of sub-cellular structures owing to their promising advantages, such as a small size with strong fluorescence, photostability, biocompatibility, and a large TP absorption cross-section. Wang et al. directly synthesized the Cu NC (Cu_14_) probe functionalized with a bifunctional peptide (Sv: CCYGGPKKKRKVG). It possessed two functional domains: (1) A clusters formation sequence (CCY), to reduce and stabilize the cluster. (2) NLS, acquired from simian virus 40 [SV40] large T antigen, which can guide the clusters inside the cellular nuclei, as shown in Figure 7A. These two domains were disconnected by a GG linker to keep the function of each domain. The Cu_14_ cluster probe emitted blue fluorescence under one-photon (OP) or TP excitation. Most importantly, as in Figure 7B, the Sv–Cu cluster probe is apt for both OP and TP cell nuclei imaging in HeLa and A549 cell lines [78].

**Figure 7 nanomaterials-12-00301-f007:**
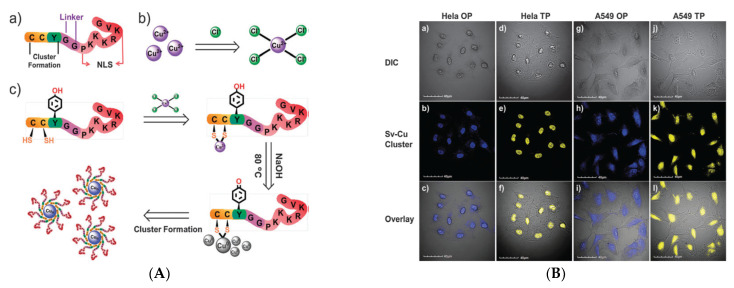
The formation of peptide–Cu NCs is depicted schematically. (**A**) the Sv bifunctional peptide, which has two functional domains. The nuclear localization sequence (NLS) was PKKKRKVG, and GG served as a linker between the two functional domains. (**B**) CuCl_4_^2−^ is formed by excess Cl^−^ and Cu^2+^ in HeLa and A549 cells. (**a**–**c**) and (**d**–**f**) are OP and TP pictures of HeLa cells, respectively; (**g**–**i**) and (**j**–**l**) are OP and TP images of A549 cells, respectively. Top panel: differentiated interference contrast, middle panel: fluorescent picture of Sv–Cu cluster (blue and yellow represent OP and TP emission, respectively), and bottom panel: combined view (bottom panel). (Reprinted with permission from ref. [72]. Copyright 2015 American Chemical Society.)

### 2.8. Computed Tomography Probe

X-ray-computed tomography (CT) is a technology that uses changes in X-ray beam absorbance in different tissues to produce a three-dimensional anatomic image that can be used to diagnose cancers, brain injury, ischemia, and other diseases [79,80,81]. CT provides the advantages of a superior spatial resolution (up to 0.5 mm), limitless penetration, cost-effectiveness, and quick 3D anatomical image acquisition. The two major limitations of CT include the limited soft tissue sensitivity and lack of functional information. The limited sensitivity of soft tissues can be mitigated by combining imaging with MRI. Despite the fact that each imaging modality has its own set of benefits, robust diagnosis may be better achieved through multi-modal applications. CT contrast agents enable strong X-ray absorption, low toxicity, and NIR fluorescence, characteristics that allow them to enhance their sensitivity and selectivity, and to distinguish between normal and pathological tissues [82,83]. Liu et al. developed lysozyme-stabilized Au NCs (Lys-Au NCs) using a continuous microwave heating approach, and yielded up to 19.61% with optimal stability, reduced toxicity and enhanced biocompatibility. As such, the inherent properties of Lys–Au NCs make them good candidates for NIRF/CT dual-modal bioimaging. Furthermore, Lys–Au NCs were conjugated with folic acid (FA) through an EDC/NHS coupling method to examine the targeting recognition capability of FA–Lys–Au NCs for fluorescence and cell imaging. As in Figure 8a, after a 4 h incubation period, the FA–Lys–Au NCs bypassed the HeLa cells and generated clear near-infrared fluorescence in the cytoplasm. Through a heightened binding affinity with folate receptors, the FA–Lys–Au NCs nanoprobe was substantially taken up by HeLa cells, suggesting that it might be employed as a contrast agent for targeted cellular imaging [84]. Additionally, Lys–Au NCs (200 mL, 5 mg mL1) were intravenously administered into HeLa tumor-carrying nude mice, as shown in Figure 8b. At 4 h post-injection, the bright fluorescence signal of FA–Lys–Au NCs could be observed at the tumor site (Figure 8), whereas Lys–Au NCs were absent in the tumor (Figure 8). The results demonstrate that FA–Lys–Au NCs have good targeting capacity for the in vivo imaging of folate receptor-positive tumors. Based on a survey of the current literature, CT with the use of Cu NCs has yet to be reported, though Cu NCs could be present as an more effective alternative compared to Au NCs.

## 3. Copper Nanoclusters (Cu NCs) in Disease Diagnosis: Biosensors and Bioimaging

Advances in nano-synthetic chemistry have enabled the production of copper NCs (Cu NCs) with optimal dimensions and excellent stability. Moreover, Cu NCs are utilized in multifarious biological applications, owing to their distinctive properties, including an enhanced luminous efficiency, elevated fluorescence lifetime, optimal optical and chemical stability, and substantial Stokes shift. These properties of Cu NCs facilitate their use as a novel class of fluorescent probes for optical sensing and bioimaging/labeling [85,86]. Discussed below are some of the major areas of disease biology in which these Cu NCs are used as biosensors, as well as for bioimaging.

### 3.1. Cu NCs as Biosensors for Early Detection of Diseases

The applications of Cu NCs in detecting important biomarkers of central nervous system (CNS), immune, bacteria and malignant disorders are explained in this section.

Dopamine is a neurotransmitter that regulates several physiological functions of the central nervous system [87]. Apart from the instrumental role it plays in the CNS, dopamine serves other biological functions, including the regulation of vasodilation, the inhibition of insulin secretion, the regulation of hormone release, and the decreased activity of immune cells. Elevated levels of dopamine have been associated with the onset of several types of cancers and schizophrenia, while a reduced level of dopamine is a hallmark of Parkinson’s disease and depression. Therefore, an accurate and facile detection of dopamine in biological samples such as human serum could serve as a biomarker for different diseases, and could aid in early detection and timely intervention. In this context, it has been observed that bovine serum albumin (BSA)-conjugated Cu NCs (BSA-Cu NCs) could selectively recognize dopamine molecules [88] in a manner that would increase the fluorescence intensity of BSA-Cu NCs proportionally with the concentrations of dopamine at physiological pH (pH 7.4). The fluorescence intensity of BSA-Cu NCs had a linear range of detection from 0.5 to 50 μM, with a detection limit of 0.28 μM. Moreover, in the presence of dopamine, Cu NC-carbon dots (Cu NC-CDs) nano-hybrids with dual emission, containing 3-aminophenylboronic acid (APBA)-modified CDs and red-emitting BSA-Cu NCs [89], produced electrons to transfer from CDs to dopamine, resulting in the fluorescence quenching (440 nm) of CDs, while the fluorescence of Cu NCs (640 nm was constant. Via this technique, the dopamine in human serum could be detected within a linear range of 0.1–100 μM, having a detection limit of 32 nM. Other groups [90] have achieved the rapid detection of dopamine with Cu NCs in the presence of H_2_O_2_, with a working range of 0.1–0.6 nM and a detection limit of 0.024 nM.

Histamine is another neurotransmitter released throughout the body, and plays a major role in inflammatory immune responses to invading microbes [90,91,92]. Aside from inflammatory responses, elevated levels of histamine have been detected in various cancer cell lines and multiple malignancies, in mast cell disorders and mastocytosis. Therefore, a successful and sensitive method for the detection of histamine could be used as a biomarker for several malignant and immune disorders. Histamine was shown to selectively quench the fluorescence of 2,3,5,6-tetrafluorothiophenol (TFTP)-coated Cu NCs [93]. The fluorescence quenching was caused by strong associations between the metal and the imidazole ring of histamine, facilitating electron transfer from the copper core to the imidazole, resulting in electron-poor locations on copper atoms. Under ideal circumstances, an effective monitoring limit of 0.1–10 μM was observed with a detection boundary of 60 nm.

Pertaining to the therapy of tumors, base excision repair (BER) inhibition has been proven to be a feasible method for targeting tumors that have been deprived of alternate repair routes, and uracil DNA glycosylase (UDG) is a key enzyme involved in BER. Moreover, many cancer subtypes and other diseases exhibit abnormal levels of UDG. DNA glycosylases are suitable targets for impeding the progress of oncology, neurodegenerative diseases, inflammatory disorders, and bacterial and viral infections. Therefore, an economic and rapid detection technique for UDG may facilitate the early detection of diseases, as well as accelerating the large-scale screening of suitable UDG inhibitors. It is noteworthy that a ratiometric fluorescence technique using poly(thymine)-Cu NCs and 4,6-diamidino-2-phenylindole (DAPI) as the produced signals for the detection of UDG [94], with a limit of detection of 0.05 U/L, has been successfully employed.

Pyrophosphate is generated when adenosine triphosphate (ATP) is hydrolyzed into adenosine monophosphate (AMP) during the synthesis of DNA or RNA. This pyrophosphate is converted into two phosphate ions catalyzed by the cellular enzyme pyrophosphatase. Pyrophosphatase is an important enzyme, as the accumulation thereof may remove water from crystals in joints [95,96,97] and tissues around them, resulting in the painful arthritis observed in diseases of the bone, such as calcium pyrophosphate deposition disease (CPDD). In addition, the cellular level of pyrophosphatase is also a biomarker in several cancers. Therefore, the early detection of abnormal levels of inorganic pyrophosphate (PPi) in the synovial or intercellular fluids may prevent CPDD or cancer in its incipient stages [98,99,100], necessitating a sensitive method to detect the levels of pyrophosphatase. In this regard, tannic acid (TA)-coated Cu NCs (TA-Cu NCs) have been used to measure the activity of pyrophosphatase [101]. In principle, the fluorescence of TA-Cu NCs could be neutralized by Fe^3+^ but reinstated upon later augmentation of PPi. However, in the presence of pyrophosphatase, PPi was decomposed into two phosphate ions, resulting in a subsequent fluorescence quenching. TA-Cu NCs were successfully shown to detect pyrophosphatase in human serum samples with a detection limit of 0.15 U/L. Moreover, glutathione-Cu NCs (GSH-Cu NCs) could exhibit an identical detection chemistry when the Fe^3+^ was replaced by Al^3+^ [102,103].

Amongst bacterial infections, *Staphylococcus aureus* can cause life-threatening infections in the form of meningitis bacteremia, pneumonia, endocarditis, and sepsis. Studies show that the amount of micrococcal nuclease (MNase), a non-specific DNA-degrading enzyme secreted by *Staphylococcus*, has a positive correlation with the pathogenicity of this bacterium [104]. MNase is a non-specific endonuclease that can break down double-stranded DNA (dsDNA) rich in adenine-thymidine (AT) or adenine-uracil (AU) nucleotide sequences. The fluorescence of dsDNA-Cu NCs gets neutralized with MNase because the dsDNA gets degraded by the MNase present in the sample [105]. The detection limit of this assay is 1 U/L and the range of detection is 1–50 U/L.

### 3.2. Cu NCs for Bioimaging

#### 3.2.1. In Vitro Studies

The synthesis of biological molecules-based Cu NCs has paved the way for their application in bioimaging processes to conduct fundamental studies, as well as for medical diagnosis. As in all biological studies, it is key to conduct in vitro studies before proceeding to in vivo experiments. By performing in vitro studies, important parameters such as particle toxicity, cellular uptake levels, and localization in sub-cellular or extracellular compartments may be evaluated seamlessly before progressing to animal and human models. Cu NCs are possibly internalized by the process of receptor-mediated or receptor-independent endocytosis. However, it was observed that the cluster internalization could be due to the crossing of the membrane at 4 °C, since at this temperature endocytosis is inhibited [106]. However, at 37 °C, the accumulation of Cu NCs within the cell clearly demonstrated endocytic uptake [107]. Previous data point to the nuclear membrane localization of Cu NCs; however, it was demonstrated that the cellular internalization and localization of GSH-Cu NCs depended on the cell line. For example, in HeLa and MDAMB-231 cells, the clusters were preferentially localized near the nuclear region, whereas in the case of A549 cells, the clusters were preferably distributed in the cytoplasm [71]. However, nanoparticles can be targeted to different organelles, such as mitochondria [108], lysosomes [109,110], endoplasmic reticulum [111], and Golgi apparatus [112] by rationally designed engineered nanoparticles.

HeLa cells (human cervical carcinoma cells) have been widely used to perform in vitro biological experiments due to their remarkable durability over several passages. The folate receptor (FR), a glycosylphosphatidylinositol-anchored cell membrane receptor, is upregulated on most malignant tissues, whereas its expression is restricted in healthy tissues and organs [32]. Therefore, folic acid (FA)-conjugated Cu NCs (FA-Cu NCs) could be exploited to target cancer cells, taking advantage of the high-affinity binding between the receptor (FR) and the ligand (FA). Indeed, Xia et al. [113] reported that FA-Cu NCs could be targeted to FR-positive HeLa cells, exhibiting blue fluorescence, while the FR-negative A549 lung carcinoma cells were unresponsive. Moreover, HeLa cells showed an increased fluorescence with the increasing concentrations of Cu NCs in a linear manner.

Previous studies have also shown that coating polyethylene glycol (PEG) on the surface of Cu NCs enhanced their biocompatibility. The incubation of Cu NCs with HeLa cells increased the bright red fluorescence of Cu NCs with increasing incubation time over a period of 24 h. Moreover, the Cu NCs aggregates were resistant to the adsorption of serum proteins, preserving their physio-chemical properties, and colocalized with the lysosomal GFP, indicating that these Cu NCs could persist in the harsh acidic environment of lysosomal vacuoles. The ability to maintain functional integrity at these extreme cellular conditions holds great promise for Cu NCs to be used as a valuable tool for bioimaging, in contrast to the frequently used biological fluorophores, which suffer from rapid photobleaching and reduced stability in harsh physiological conditions [114].

Another hallmark feature of cancer cells is the difference in the pH of their microenvironment [115,116]. Due to their heavy reliance on glycolysis, cancer cells produce higher levels of lactic acid, and this released acid makes the cancer environment more acidic. However, the accepted notion is that the intracellular pH of cancer cells is slightly alkaline as compared to the corresponding normal tissues. Hence, the conception of probes that are able to monitor pH alterations is especially crucial in detecting cancer at nascent stages. Cu NCs functionalized with cysteine/chitosan exhibited good pH responsiveness (pH 4.5–8; red fluorescence at acidic pH and blue-green to green florescence at alkaline pH) [117] and could be internalized by breast adenocarcinoma (MCF-7) and human embryonic kidney (HEK-293) cells. A 40 min incubation of cysteine/chitosan-Cu NCs with MCF-7 cells demonstrated a transition from red to bright green fluorescence, indicating an alkaline environment, and the HEK cells also exhibited similar properties, albeit with less fluorescence shift. Following the embodiment into living cells, the Cu NCs demonstrated a cellular pH environment-dependent luminescence shift with orange-red emission at pH 4.5, while a bright green emission was seen over a temporal frame at pH 7.4, through their aggregation-induced emission (AIE) kinetics. These results provide a platform for using the Cu NCs probes to distinguish between cancerous and healthy tissues, and may facilitate the early diagnosis of cancers, resulting in timely interventions.

Calcium ions (Ca^2+^) play a major role in cellular signaling pathways, as well as neuronal death resulting in the onset of several neurodegenerative diseases. Therefore, the development of a sensitive fluorescent technique enabling Ca^2+^ sensing and imaging could ease the detection of Ca^2+^ in neurodegenerative disease. A recent study reported that Cu NC-based ratiometric fluorescent probes could be used for the real-time biosensing and imaging of Ca^2+^ in neuronal cells [63]. The probe was modified with a distinct Ca^2+^ ligand containing two formaldehyde groups and coupled with polyethyleneimine (PEI) to result in a novel ligand molecule for the production of Cu NCs. The fluorescent nanoprobe demonstrated a linear range of assay sensitivity with a Ca^2+^ concentration of 2–350 μM, and a detection limit of about 220 nM. Furthermore, this ratiometric probe exhibited high specificity, elevated stability, low toxicity and excellent biocompatibility. This study also enhanced our understanding of superoxide-induced neuronal death caused by calcium overload at various parts of neurons. This study not only developed an elegant technique to image Ca^2+^ in neurons to detect neurodegenerative diseases, but also reported a fundamental calcium signaling pathway.

#### 3.2.2. In Vivo Studies

The benefits of Cu NCs for in-vivo imaging are due to their high cell membrane penetration, high luminescence, greater stability, low toxicity, and low tissue absorption [118,119]. Cu NCs are excellent biocompatible molecules as they are amenable to modifications by capping with non-toxic biocompatible macromolecules, such as polysaccharides or proteins. Most of the studies concerning Cu NCs describe no or low cytotoxicity, independent of the size and the surface ligand composition. However, one study [120] showed that BSA-Cu NCs trigger muscle cell apoptosis and atrophy of C2C12 myotubes in a dose-dependent manner, with cell viability decreasing by 88% at 50 µg/mL of Cu NCs concentration. The Cu NCs induced the proapoptotic cellular death pathway with increased expression and activity of Bax/Bcl-2 and caspase-3/9, respectively. Moreover, Ser/Thr Akt kinase-mediated myotube atrophy was stimulated followed by the transcription of atrophy-related genes [120]. Understanding the mechanism of nanocluster toxicity will enable strategies to better design clusters with reduced toxicity. The redesigned strategies will need to consider the mode of toxicity without compromising its ability to perform its desired function. For reduced cell surface interactions, nanomaterials can be designed to have negative surface charge or be combined with polyethylene glycol to reduce protein binding. The nanoparticle can be capped with a shell material to reduce dissolution into toxic ions, or a chelating agent can be combined onto the surface. To decrease oxidative stress, the band gap of the material can be tuned by doping, or antioxidant molecules can be tethered to the nanoparticle surface [121].

As mentioned earlier, PET imaging is a powerful diagnostic technique, especially for the in vivo detection of tumour growth. PET is characterized by high sensitivity, temporal precision, and appreciable tissue traversal. The creation of probes that can aim at specific receptors or molecules in tumour cells is extremely valuable for clinically relevant PET imaging. However, the use of the radioactive copper (^64^Cu) as a PET imaging probe, coupled to biomolecules through macrocyclic chelators, may lead to misleading results because of the separation and transchelation of ^64^Cu. Other biochemical issues that may hamper this technique from being successfully used in clinical settings include toxicity, the unreliable detection of budding malignant tissues, and an inaccurate diagnosis of cancer markers. Addressing these factors is very important to making use of this promising bioimaging tool for the benefit of human health. The targeted Cu NCs were reported by resolving these issues, rendering them highly capable to be translated from the bench to the bedside [68]. The very small structure of Cu NCs enables efficient system clearance, reduced off-target effects, and selective tumour targeting in an in vivo mouse model. Innate radiolabeling via the explicit addition of ^64^Cu to Cu NC conjugated with FC131 peptide (^64^Cu-CuNCs-FC131) provided a high specific activity for sensitivity and precision in detection via high-affinity binding with FC131-receptor (CXCR4)-positive triple negative breast cancer (TNBC) patient-derived xenograft mouse models and human TNBC tissues. In addition to revealing the promise of CXCR4-targeted ^64^Cu-CuNCs to be used in TNBC clinical imaging, this study also provided an advantageous method to produce and evaluate the potential of nanoparticles to be applied as cancer theranostics.

A similar strategy was employed in an in vivo mouse model of cancer. In this study, ultrasmall chelator-free ^64^Cu nanoclusters utilizing bovine serum albumin (BSA) as a scaffold for PET imaging in an orthotopic lung cancer model were developed [67]. The authors targeted the luteinizing hormone releasing hormone receptor (LHRHR) in tumour tissues, and its high-affinity ligand LHRH was coupled to BSA molecules to assemble [^64^Cu]CuNC-BSA-LHRH. This Cu NCs demonstrated a high radiolabeling stability, ultrasmall size, and specific tumour targeting, as well as robust renal clearance. PET imaging using ^64^CuNCs as the probe demonstrated a more precise, accurate, and deeper-penetration imaging of orthotopic lung cancer in vivo as opposed to near-infrared fluorescence imaging. These results are highly encouraging in the sensitive detection of tumour tissues in clinical settings.

Apart from PET imaging, Cu NCs have also been favorably used with MRI modalities. A recent study by Zhang and colleagues exploited the functionality of MRI to administer radiolabeled Cu NCs to diffuse intrinsic pontine glioma by virtue of a focused ultrasound-mediated opening of the blood–brain barrier (BBB). Since the BBB is rather intact in this case of glioma, it is a challenging task to deliver diagnostics or therapeutics to the target, and Cu NCs are no exception. However, following the MR-guided focused ultrasound opening of the BBB, radiolabeled Cu NCs were effectively delivered to the pons of naïve mice. The radiolabeled Cu NCs were observed to display considerable tumour uptake, and were well-distributed and captured by the tumor [122].

## 4. Summary and Future Outlook on Clinical Translation

In this review, we have highlighted the advancements made in biosensing, and in vitro and in vivo bioimaging, using Cu NCs as efficient probes. Though the technique holds great promise in biomedical applications, it is at present limited in its scope for use in relevant clinical settings. First, the corona effect of Cu NCs limits its use and requires more systematic studies to resolve this issue. Secondly, Cu NCs are prone to oxidation, requiring controlled synthesis, thereby limiting their use as a widespread probe in biological systems. Thirdly, Cu NCs show low QY, leading to difficulties in analyte detection. However, the aggregation-inducing effect can profoundly improve the QY of Cu NCs, though it must be considered that the downside is that the same effect could be non-specifically brought about by the influence of the surrounding microenvironment of biological samples, causing difficulties in the interpretation of results. Although the formation of self-assembled aggregates of highly compact and well-ordered Cu NCs can effectively increase the emission intensity, stability, and tunability, their synthesis demands organic solvents, which are typically incompatible for use in biological applications. These encumbrances lead to reduced imaging accuracy, decreased drug delivery to the target (reducing drug efficiency), and toxicity caused by non-specific retention. These are some of the challenges that have limited the applications of Cu NCs in in vivo bioimaging studies. However, the most promising feature of these in vivo studies is that mice and humans share metabolic homogeneity, having analogous organ structures and systemic physiology, and they also exhibit considerable similarities in disease pathogenesis. For example, mouse tumors have histological and biochemical features that are highly similar to human tumours. Therefore, these studies on mouse models could be extrapolated to humans more conveniently than anticipated. However, it goes without saying that a lot of thoughtful research must be put in to reap the benefits of using the Cu NCs as biosensors and imaging probes in human health and disease.

Although the effective emission properties of Cu NCs could be manipulated by designing the dispersion medium, this is not useful in clinical settings, because water is the medium of choice for most of the drug administration studies at the preclinical and clinical stages. The substitution of water with any organic solvent that might increase the efficacy of Cu NCs may cause acute or chronic toxicity in vivo. Moreover, the contamination of dispersion medium with extracellular and bodily fluids cannot be avoided, resulting in a possible change in the properties of the dispersion medium, and therefore, probable reductions in the efficacy of the Cu NC. More investigations on the impact of these bodily fluids on the emissive properties of Cu NCs are, therefore, instrumental to transcend the clinical endpoints of CuNC-based theranostic agents.

To promote the clinical translation of Cu NCs, methods of synthesizing and engineering Cu NCs must be further advanced so that the chemical, physical, and biological properties of the clusters can be more effectively controlled. On a more hopeful note, the rapid progress in the chemistry and engineering of Cu NCs over the years has reduced the gaps between research and practice. Exploiting the strong body of literature already established by studies on other metal NCs (e.g., Au and Ag NCs) in disease diagnosis and treatment, it is reasonable to predict that insights derived from these studies will lead to the development of Cu NCs as useful clinical reagents in the near future [10].

## 5. Conclusions

We have detailed the various applications and modes under which Cu NCs may be adopted as probes for disease detection. The functionalization of Cu NCs with magnetic, fluorescent and radioactive materials may render them beneficial for use with magnetic-, optical- and nuclear-imaging modalities, respectively. Multiple factors, including reaction temperature, pH, ligand and cluster size, are capable of influencing the final synthesized product, and the precise monitoring and controlling of these factors are instrumental in deriving Cu NCs of the optimal size and targeting capabilities. This in turn plays a key role in biosensing and bioimaging applications, whereby the NCs serve the function of identifying molecular targets during the initial stages of disease, facilitating early diagnoses.

## Figures and Tables

**Figure 2 nanomaterials-12-00301-f002:**
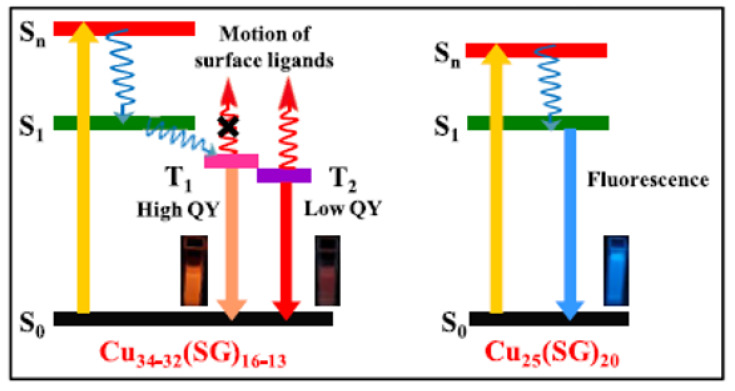
Schematic illustration of radiative decay dynamics and the origin of emission in Cu_34−32_(SG)_16−13_ and Cu_25_(SG)_20_ NCs. (Reprinted with permission from ref. [40]. Copyright 2019 Asian Chemical Editorial Society.)

**Figure 4 nanomaterials-12-00301-f004:**
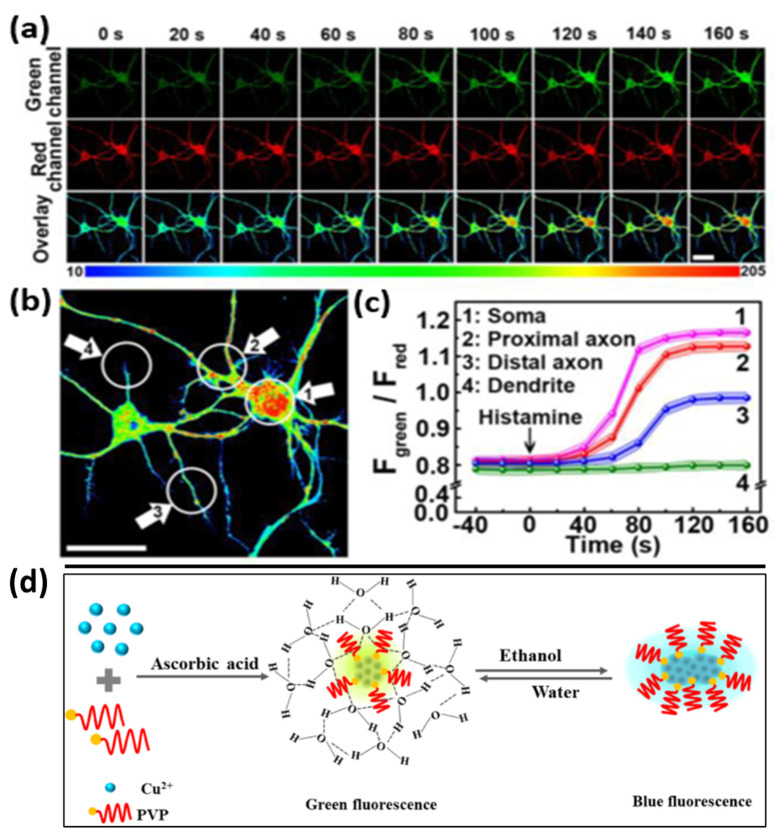
(**a**) A histamine concentration 50 μM was used to promote the time-tracking of confocal fluorescence microscopic images of neurons. (Reprinted with permission from ref. [63]. Copyright 2019 American Chemical Society.) (**b**) An expanded picture of neurons activated for 160 s by histamine. The soma, proximal axon, distal axon, and dendrite of a neuron are represented by the different zones (14) accordingly. (**c**) F_green_/F_red_ values of distinct zones in neurons after stimulation with 50 M histamine for various periods of time. The average fluorescence intensity obtained from the green channel (510,650 nm) and red channel (660,780 nm) is represented by F_green_ and F_red_, respectively. The wavelength of excitation was 488 nm. F_red_ = F_660780_, F_green_ = F_510650_. Scale bars = 25 μm. (**d**) The synthesis and fluorescence adjusting of PVP-Cu NCs are depicted schematically. (Reprinted with permission from ref. [64]. Copyright 2021 Talanta.)

**Figure 5 nanomaterials-12-00301-f005:**
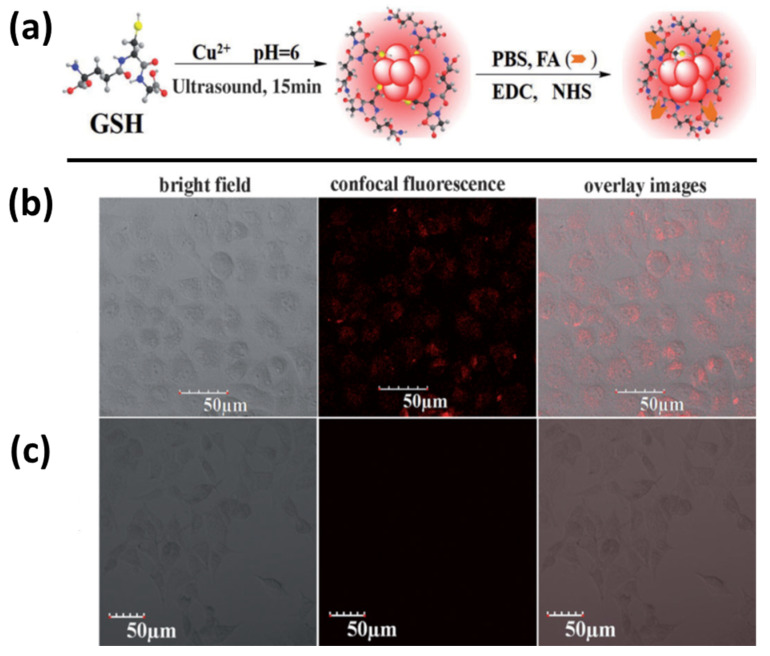
(**a**) Typical sonochemical production of red fluorescent GSH-stabilized Cu NCs and folic acid bioconjugated to GSH-Cu NCs. (**b**) HeLa cells and (**c**) 293T cells incubated with FA-conjugated GSH-Cu NCs images captured by laser scanning confocal microscopy. (Reprinted with permission from ref. [65]. Copyright 2015 Asian Chemical Editorial Society.)

**Figure 6 nanomaterials-12-00301-f006:**
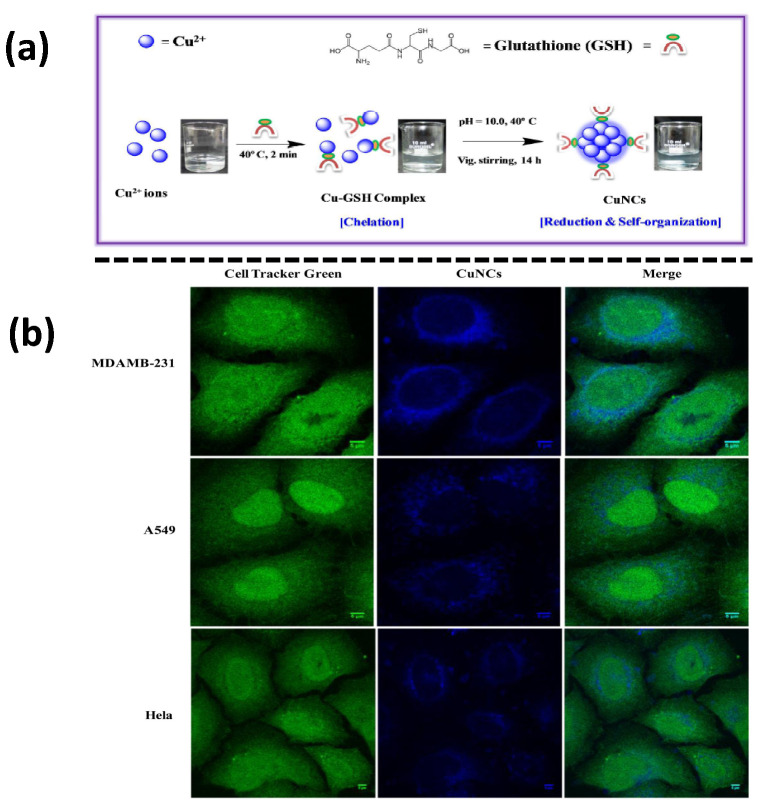
(**a**) With GSH as a template, the formation of blue-emitting Cu NCs was stabilized, as were the associated digital pictures. (**b**) Cu NC subcellular localization in HeLa, A549, and MDAMB-231 cells after 12 h of incubation at 37 °C. Before being fixed in 4% paraformaldehyde, the cells were labeled with Cell Tracker Green CMFDA (5-chloromethylfluorescein diacetate). A Laser Scanning Confocal Microscope was used to capture images of the fixed cells. The scale bar of the images reflects 5 μM in spatial size. (Reprinted with permission from ref. [71]. Copyright 2015 American Chemical Society.)

**Figure 8 nanomaterials-12-00301-f008:**
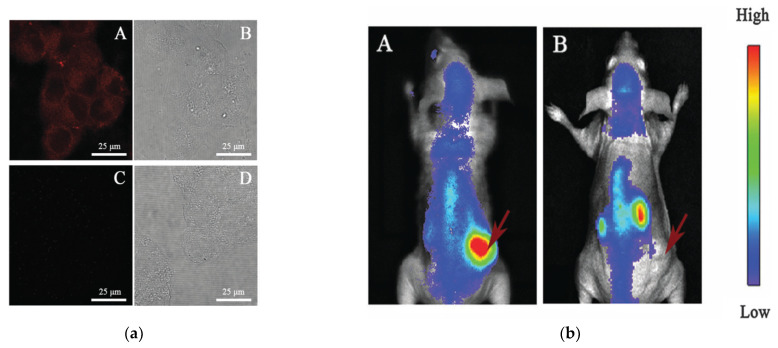
(**a**) Confocal microscopy observations of HeLa cells treated for 4 h with FA–Lys–Au NCs (**A**) and Lys–Au NCs (**C**); (**B,D**) the matching bright-field images. (**b**) At 4 h after injection with (**A**) FA–Lys–Au NCs and (**B**) Lys–Au NCs via the tail vein, in vivo fluorescence imaging of HeLa tumor-bearing nude mice. The tumor is shown by the brown arrow. (Reprinted with permission from ref. [84]. Copyright 2016 Royal Society of Chemistry.)

## Data Availability

Not applicable.

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
