# Peer review of "The Multifarious Applications of Copper Nanoclusters in Biosensing and Bioimaging and Their Translational Role in Early Disease Detection"

_nanomaterials, 2022, doi:10.3390/nano12030301_

Round 1
Reviewer 1 Report
This is a review article, in which the development of copper nanoclusters (Cu NCs) as imaging probes in different imaging modes was reviewed, and their applications in biosensing and bioimaging were introduced. This review might be eventually published, but a major revision is needed.
(1) The references published in recent two years (2020 and 2021) are rare, and some are not relevant with Cu NCs. Some recent research progress should be added.
(2) In 2.3 section, the authors introduced that Cu NCs could be used as MRI probe. So, corresponding MRI results should be given in Figure 5, but not only FI results.
(3) 2.7 section should be deleted since only the examples of Au NCs were given, which is not consistent with the topic of this review.
(4) Corresponding references should be given in the caption of each figure.
(5) A clearer Figure 1 should be given.
(6) Please note Superscript and subscript, for examples, NaBH4 and Cu2+ in the caption of Figure 1; Cu25(SG)20 NCs , Cu34–32(SG)16-13 NCs and S2- in the caption of Figure 3, Fred = F660780, Fgreen = F510650 in the caption of Figure 4, CuCl42-, Cl-, Cu2+ in the caption of Figure 7. NH4+ in the main test.
(7) The full name of some abbreviates should be given. For example, GOx enzyme.
Reviewer 2 Report
In this review, Busi et al. have reviewed the recent trends and applications of copper nanoclusters (Cu NCs). This is an important topic but a lot of mechanisms are not clear in its current format. I suggest major revisions to reconsider this paper to be accepted for publication in Nanomaterials.
- Abstract - needs to be rewritten, along with challenges in exiting technologies. authors never mentioned ths forms of Cu nanclusters, for example, which forms of clusters they have reviewed? and why did they choose Cu over other metallic nanoparticles such as gold and silver as both of them give good results in terms of detection. Cu has also been used as therapy agent, so how authors feel this is a great choice in terms of detection agent. This should clearly be discussed within the manuscript too.
- 'The early diagnosis of diseases is an important but challenging task, especially in diseases that do not manifest telltale signs and symptoms until they have progressed to a more advanced stage' which diseases, please give examples. also this sentence needs reference; authors should cite this paper. https://doi.org/10.1016/j.coelec.2021.100786
- advantages and disadvantages of existing technologies?
- why copper over other metals?
- Importance of optical windows in fluorescence, such as UV/Vis, NIR-I, NIR-II. why these windows are important along with limitations and solutions.
- 'In biological organisms, red or infrared (IR) fluorescence may be utilized to avoid interference with the innate fluorescence of living species as compared to many luminous compounds deliver other visible colour such as blue or green.' this sentence needs reference, https://doi.org/10.1063/5.0065833
- Authors keep switching between QY and quantum yield. please make it consistent.
- Effect of ligand, any effect of PEG, PVP etc? or other polymers or dyes?
- In most of the figures, authors never gave a statement on copy right permissions. Surprisingly, in few figures references are not given either.
- 'Liu et al. reported that a Cu NC-based fluorescent probe was advanced for real-time biosensing and bioimaging of cytoplasmic Ca2+ in neuronal cells'. no reference for this?
- Authors keep switching between MRI and magnetic resonance imaging, please make it consistent.
- In terms of bioimaging, detailed description of uptake/cellular localisation, internalisation is required. In depth description of subcellular localisation is also required, for example where these particles end up in cells, e.g., mitochondria, nucleus, https://doi.org/10.1016/j.bbiosy.2021.100023
- Toxicity (on-target, off-target), biocompatibility, of these particles?
- how toxicity can be minimised?
- what are clinical endpoints? and what are translational barriers?
- how this technology is better than other technologies?
- Raman imaging modality is missing as well. please highlight this too.
Round 2
Reviewer 1 Report
The authors have revised the manuscript as suggestion. The revised manuscript can be accepted now.
Reviewer 2 Report
I am pleased to recommend the revised manuscript for publication in Nanomaterials.